# Understanding the combined symptom medication score in the light of contexts and mechanisms

Anne Møller Lohmann[1,2*], Anne Poder Petersen[2,3], Johannes Martin Schmid[3], Hans Jürgen Hoffmann[2], Jeanette Finderup[2,4]

**1** Department of Public Health, Aarhus University, Aarhus, Denmark, **2** Department of Clinical Medicine, Aarhus University, Aarhus, Denmark, **3** Department of Respiratory Diseases and Allergy, Aarhus University Hospital, Aarhus, Denmark, **4** Department of Renal Diseases, Aarhus University Hospital, Aarhus, Denmark

* anneml@clin.au.dk

## Abstract

### Background

In a clinical trial of allergen-specific immunotherapy for allergic rhinoconjunctivitis, the Combined Symptom Medication Score (CSMS) was utilized as the primary endpoint. This was aligned with the European Academy of Allergy and Clinical Immunology recommendation. However, participants wanted to elaborate on how their behaviour affected their score, so voluntary free text boxes were added to the CSMS questionnaire. This study aimed to evaluate the patient-reported outcomes registered in the free text boxes to identify and understand contexts and mechanisms that may affect the CSMS.

### Methods

The realist evaluation methodology was followed in four iterative steps: 1) development of the initial programme theory and context-mechanism-outcome (CMO) configurations, 2) collection of evidence, 3) data analysis, and 4) interpretation and assessment of results.

### Results

Seven CMO configurations were identified, highlighting contexts and mechanisms that may affect the CSMS. These included CMO1 – decision on preventive relief medication dosage, CMO2 – exhibiting symptom-relieving behaviour, CMO3 – being exposed to different levels of grass pollen, CMO4 – mistaking other symptoms for grass pollen-induced symptoms, CMO5 – different exposure to grass pollen when travelling abroad, CMO6 – reporting of relief medication for other allergies in CSMS, and CMO7 – failure to report symptoms not included in CSMS.

**Data availability statement:** The data supporting the findings of this study contain potentially sensitive health-related information and personally identifiable comments from free text boxes. As a result, data sharing is subject to ethical and legal restrictions. Therefore, data access is restricted by the guidelines set by The Scientific Ethical Committees for Region Midtjylland, which has imposed these restrictions to protect participant confidentiality. Data requests can be submitted to The Scientific Ethical Committees for Region Midtjylland at: Skottenborg 26, DK-8800 Viborg, Denmark Email: komite@rm.dk Website: www.komite.rm.dk The authors do not have sole authority over data access decisions.

**Funding:** The author(s) received no specific funding for this work.

**Competing interests:** The authors have declared that no competing interests exist.

## Conclusion

This realist evaluation contributes verified CMO configurations based on patient perspectives to understand how context and mechanisms may affect the CSMS. We recommend further investigation in quantitative studies, as awareness of these CMOs may increase the internal validity of future allergy trials using CSMS as an endpoint.

## Trial registration

ILIT.NU: EudraCT 2020-001060-28.

## Introduction

Allergic rhinitis (AR) and allergic rhinoconjunctivitis (ARC) are increasingly recognized public health issues [1,2]. AR and ARC affect general well-being by reducing the quality of life and sleep, and direct and indirect costs are associated with managing these diseases [1]. Worldwide, 40% of the population is affected by AR, while ARC affects 10–30% of adults and up to 40% of children [3]. A longitudinal cohort study conducted in Denmark followed 276 children from birth to 26 years of age and found that allergic diseases occur not only in childhood but persist into adulthood. Furthermore, the cumulative prevalence of ARC from birth to 26 years was 27.9% [4]. AR and ARC are thus prevalent diseases that affect public health, the economy, and the individual's well-being [1,2]. Therefore, working towards effective prevention and treatment strategies is vital to improve overall health.

ARC is typically treated with symptomatic medication [3]. Although allergen immunotherapy is available for severe cases, it poses challenges due to its three-year duration, associated side effects, and high cost [5]. Intralymphatic immunotherapy (ILIT) has emerged as a promising alternative, offering a shorter, more convenient, and effective treatment option [6]. The ILIT.NU trial, EudraCT No. 2020-001060-28, has enrolled 516 participants in an international multicentre study to investigate the efficacy of ILIT. There were sites in Denmark, Switzerland, and Sweden. Data for this realist evaluation were collected from the Danish sites. Patients were randomized 2:1 to verum or placebo. The grass pollen season of 2021 served as the baseline year, while the seasons of 2022 and 2023 served as follow-up after the intervention injections. The primary endpoint was the Combined Symptom Medication Score (CSMS), following the European Academy of Allergy and Clinical Immunology (EAACI) recommendation in its 2014 position paper [7]. Over the three years, participants were required to answer the CSMS questionnaire daily during the Danish grass pollen season, which lasts from May to August. The CSMS is structured for respondents to score nasal symptoms (itchy nose, sneezing, runny nose, and blocked nose) and conjunctival symptoms (itchy/red eyes and watery eyes) on a scale of 0–3 [7]. Respondents also note their medication, where oral and/or topical (eyes or nose) non-sedative H1 antihistamines score 1, intranasal corticosteroids score 2, and oral corticosteroids score 3 [7]. During the 2023 season, participants answered the updated CSMS. Fig 1 illustrates the original CSMS with the added ad hoc questions (the updated CSMS).

The position paper describes the development of the CSMS. However, whether the development included pilot and field testing, as recommended in the 'COnsensus-based Standards for the selection of health Measurement INstruments' (COSMIN) taxonomy, is unclear [8,9]. According to the COSMIN taxonomy, involving the target population in the pilot and field-testing phases is essential before validation. This is important in this instance, as only participants with ARC can accurately judge the comprehensibility, relevance, and completeness of the CSMS [8,9]. Even though it is unclear whether participants were involved in the development phase, we have identified two validation studies of the CSMS [10,11]. They both analyzed the correlation between the CSMS and existing validated questionnaires. One of the studies was by Sousa-Pinto et al., who analyzed several CSMS questionnaires that included scales different to the original recommended in the position paper. Therefore, the CSMS utilized in the ILIT.NU trial was not entirely comparable to these [11]. Both

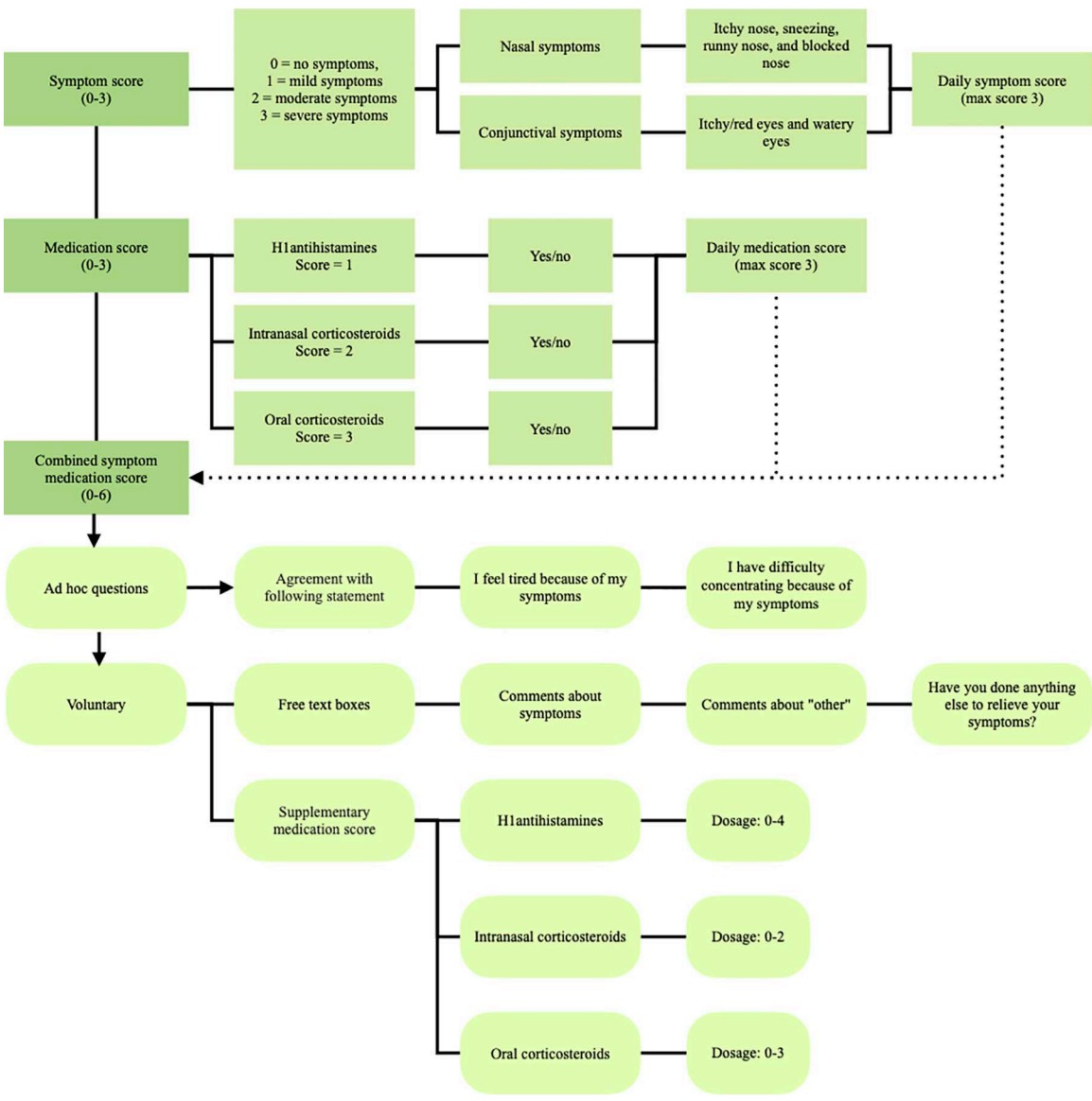

**Fig 1. Illustration of CSMS.**

of the identified validation studies concluded that the CSMS is a possible primary endpoint for clinical allergen-specific immunotherapy trials for ARC. Nonetheless, the authors emphasize the need for further validation and evaluation of the CSMS [10,11]. Therefore, the CSMS that is used in the ILIT.NU trial is still not wholly validated, and it is unclear whether the perspectives of participants with ARC were included in the original development, which is why its validity and reliability may be open to question [8,12,13].

In the ILIT.NU trial, participants with ARC were involved in the research process to improve the research design [14,15]. Initially, two focus groups were conducted to explore ways to enhance the quality and quantity of the patient-reported outcome data of the CSMS, thereby increasing retention in the ILIT.NU trial [16]. Results from these focus groups indicated that participants were highly interested in providing accurate data, driven by a strong commitment to supporting research and the hope that the treatment would ultimately benefit both themselves and others. Therefore, they wanted to provide more detailed information about how their behaviour affected their symptoms and medication scores than the original CSMS allowed. Consequently, voluntary free text boxes and voluntary registration of medication dosage were added to an updated version of the CSMS as ad hoc questions, enabling the participants to elaborate on their responses. Furthermore, three additional ad hoc questions were added to the updated version. These questions were based on the literature and addressed fatigue, difficulties concentrating, and the use of any additional medications to relieve symptoms [17–19]. When analyzing the difference between treatment groups in the ILIT.NU trial, the original CSMS could not detect a significant difference. However, the updated CSMS showed a significantly lower use of additional medication among participants treated with three successful lymph node injections compared to controls [20]. This finding emphasizes the importance of incorporating the patient perspective into research [21]. Therefore, the information gathered from the free text boxes in the ILIT.NU trial presents an opportunity to understand how the behaviour of the participants affects the CSMS to determine what works and under what circumstances. This can be achieved by conducting a realist evaluation, as this method provides a comprehensive understanding of an intervention's effect and can be used to identify the contexts and mechanisms that may affect the CSMS [22,23]. Therefore, this study aimed to conduct a realist evaluation to identify and understand the contexts and mechanisms that may affect the CSMS collected through free text boxes in the ILIT.NU trial.

## Methods

### Study design

A realist evaluation involves analyzing an intervention's underlying contexts and mechanisms to determine what works, for whom, in what circumstances, and how [22–24]. This is operationalized through context-mechanism-outcome (CMO) configurations to identify the different contexts that activate specific mechanisms to achieve effects (outcomes). Therefore, this realist evaluation included four iterative steps: 1) development of the initial programme theory and CMO configurations, 2) collection of evidence from the ILIT.NU trial, 3) data analysis, and 4) interpretation and assessment of results [22,25]. Data analysis was managed using NVivo 14, Excel, and Stata/MP 18.0. Demographic and baseline characteristics were analyzed by descriptive statistics where statistical significance was determined with a chi-square test comparing the group that provided comments to the group that did not provide comments. Fig. 2 presents a flow diagram detailing the periods during which participants were required to complete the CSMS in the ILIT.NU trial, the timing of the focus groups, and the timing of the realist evaluation. It also illustrates the steps when data from the focus groups and the free text boxes informed the realist evaluation. Participants for the ILIT.NU trial were recruited from 22/03/2021–14/06/2021.

**Step 1: Development of the initial programme theory and CMO configurations.** The initial programme theory explains mechanisms and contexts influencing the outcomes of interest (in this case the CSMS) [22,25]. Firstly, the data from the focus groups were examined and coded inductively with a focus on identifying the mechanisms and contexts that might affect the CSMS. The final preliminary themes reflected situations that could increase or decrease symptoms or

medication in a manner that was not represented in the CSMS or inaccurately affected the CSMS. The identified themes were recognized and consistent across the focus groups. For example, one participant explained:

> "I think there were some days where my symptoms, for obvious reasons, were non-existent. Like, either it was one of those days where it was just really raining, or I had been indoors, or I had… you know, one of those times where you think to yourself, 'I've had three days without symptoms,' but that's not really true in reality."

This type of reflection contributed to the development of a theme related to variations in exposure. The quote illustrates how daily routines and environmental factors, such as weather or time spent indoors, affect allergen exposure and symptom reporting, potentially leading to an overestimation or underestimation of symptom burden in the CSMS. Based on these themes, the initial programme theory and CMO configurations were developed [22,25].

**Step 2: Collection of evidence from the ILIT.NU trial.** Qualitative data from the free text boxes in the updated CSMS and participant demographic characteristics were collected as evidence.

**Step 3: Data analysis.** The first phase of the analysis involved coding the qualitative data [23,24]. As in step 1, the data were coded into themes, but this time, some of the themes were refined, and additional themes were found. The codes and categories were then matched with the participants, and this frequency was tailed.

This step resulted in verified CMO configurations through comments from the free text boxes.

**Step 4: Interpretation and assessment of results.** The last step involved interpreting and assessing the analysis, during which the CMO configurations were further refined, resulting in a refined programme theory [22,25].

## Ethical approval

The ILIT.NU trial received ethical approval from the Central Denmark Region Committee for Ethics (Identifier: 1-10-72-315-20). All participants received oral and written information about the project and provided written informed consent before inclusion, which could be withdrawn at any time. The recommendations for data storage from the General Data Protection Regulation (GDPR) and the Danish Data Protection Act were followed [26]. This realist evaluation did not require further approval.

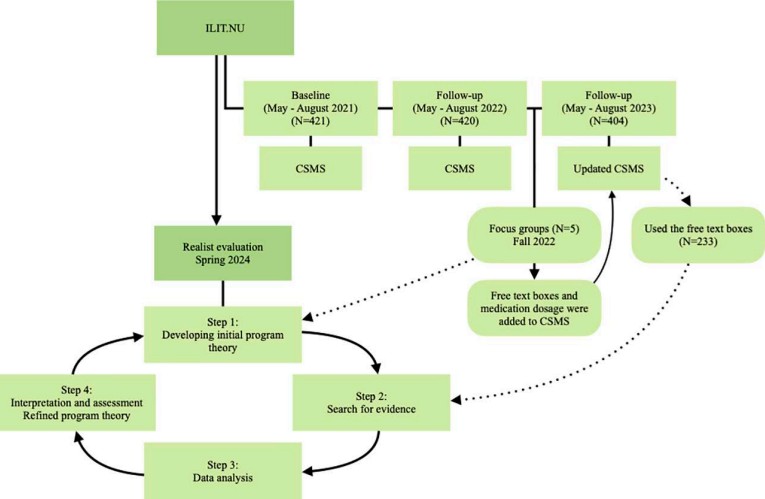

**Fig 2. Flow diagram.**

## Results

### Step 1: Development of the initial programme theory and CMO configurations

Five preliminary themes were identified: medication, behaviour, exposure, symptoms, and vacation. The initial CMO configurations based on these are represented in Table 1 as the initial programme theory.

### Step 2: Collection of evidence from the ILIT.NU trial

A total of 233 out of 404 participants made comments in the free text boxes. The study population for this evaluation consists of the 233 participants who provided comments, and this population provided a total of 2159 comments. Table 2 presents the demographic characteristics of Danish participants in the ILIT.NU trial, categorized based on whether they provided comments in the free text boxes. More than half of them provided comments (58% versus 42%). Additionally, more females than males provided comments (59% versus 41%, $P = 0.001$). In total, 69% of the participants who provided comments were in the treatment group, and 31% were in the placebo group, which is aligned with the overall distribution

**Table 1. Initial programme theory.**

| Initial programme theory | | | | | | |
|---|---|---|---|---|---|---|
| | **Context** | | **Mechanism** | | | **Outcome** |
| **Initial CMO1** | When the participants plan their day | → | they choose to take their relief medication **before experiencing symptoms** | → | | which impacts their symptoms |
| **Initial CMO2** | When the participants experience allergy symptoms | → | it affects their behaviour as they **try to decrease the symptoms** | → | | which results in fewer symptoms |
| **Initial CMO3** | When the participants move about in their daily life | → | it will affect their **exposure** | → | | which determines the degree of their symptoms |
| **Initial CMO4** | When the participants assess their symptoms | → | they may mistake the allergy symptoms for **other symptoms** | → | | which leads to overre-porting of the symptoms |
| **Initial CMO5** | When participants **travel abroad** | → | their **exposure to pollen** differs from the national pollen count | → | | which can affect their symptoms |

**Table 2. Demographic characteristics.**

| | Patients | | | | | | | | | | | |
|---|---|---|---|---|---|---|---|---|---|---|---|---|
| | **Total** *n (%)* | | | | **Treatment** *n (%)* | | | | **Placebo** *n (%)* | | | |
| | Total | Comments | No comments | *P-values* | Total | Comments | No comments | *P-values* | Total | Comments | No comments | *P-values* |
| *n* | 404 (100) | 233 (100) | 171 (100) | | 275 (68) | 161 (69) | 114 (67) | | 129 (32) | 72 (31) | 57 (33) | |
| **Age** | | | | | | | | | | | | |
| 18–24 | 120 (30) | 69 (30) | 51 (30) | | 80 (29) | 46 (29) | 34 (30) | | 40 (31) | 23 (32) | 17 (30) | |
| 25–31 | 115 (28) | 64 (27) | 51 (30) | | 90 (33) | 52 (32) | 38 (33) | | 25 (19) | 12 (16) | 13 (23) | |
| 32–38 | 63 (16) | 35 (15) | 28 (16) | | 36 (13) | 20 (12) | 16 (14) | | 27 (21) | 15 (21) | 12 (21) | |
| 39–45 | 65 (16) | 41 (17) | 24 (14) | | 44 (16) | 26 (16) | 18 (16) | | 21 (16) | 15 (21) | 6 (11) | |
| 46–52 | 22 (5) | 13 (6) | 9 (5) | | 12 (4) | 8 (5) | 4 (4) | | 10 (8) | 5 (7) | 5 (9) | |
| 53–62 | 19 (5) | 12 (5) | 8 (5) | | 13 (5) | 10 (6) | 4 (4) | | 6 (5) | 2 (3) | 4 (6) | |
| | | | | *0.713* | | | | *0.951* | | | | *0.166* |
| **Gender** | | | | | | | | | | | | |
| Female | 208 (51) | 137 (59) | 71 (42) | | 145 (53) | 97 (60) | 48 (42) | | 63 (49) | 40 (56) | 23 (40) | |
| | | | | *0.001* | | | | *0.003* | | | | *0.086* |

of participants in the treatment and placebo groups, where the participants were randomized 2:1. The age of the participants was not associated with the likelihood of providing comments in the free text boxes.

## Steps 3 and 4: Data analysis, and interpretation and assessment of results

The analysis expanded the initial five CMO configurations for grass pollen-induced ARC to seven CMOs which are illustrated in Table 3 as the refined programme theory.

Table 4 presents the distribution of participants' contributions to various CMOs and the median for times they contributed, categorized based on whether they were in the treatment or placebo group. The most common CMOs are CMO3, CMO5, and CMO7. Overall, response frequencies were not different between the treatment and placebo groups.

**CMO1 – Preventive relief medication.** In the ILIT.NU trial, participants were requested to refrain from taking preventive relief medication to better observe changes in symptoms. However, 15 participants reported taking preventive relief medication at least once, as they explained it could alleviate their symptoms. In the free text box, one participant stated, "I took the medicine preventively because I had to mow the grass and be outside all day", while another noted, "I had to be outdoors for a few hours, so I took a pill to avoid symptoms". A result of this could be underreporting of symptoms and, conversely, overreporting of the use of medication.

**CMO2 – Behaviour.** Patients do not challenge their allergy when they try to avoid being exposed to grass pollen, as illustrated in the following quotations: "I have stayed indoors all day to minimise my symptoms. It has been a good break!" and "Been inside all day to avoid pollen". The participants' behaviour may affect the CSMS if they experience fewer symptoms and use less medication, leading to underreporting of both.

**CMO3 – Exposure.** This CMO configuration was supported by comments about being either inside or outside. As almost half of the participants commented about being inside or outside, it appeared to be an important concern for them. If a participant stays inside for the entire day, they may not require any medication nor experience symptoms. This is supported by the following quotations: "Have been at home and mostly inside, where there is not much pollen" and "Worked inside all day". On the other hand, being outside for extended periods can exacerbate symptoms of grass pollen-induced ARC and the need for medication, as another participant noted: "Great pollen impact due to outdoor dog training". Therefore, an individual's location and exposure level seemed to have an important impact on the CSMS.

**Table 3. Refined programme theory.**

| | Refined programme theory | | | | |
|---|---|---|---|---|---|
| | Context | | Mechanism | | Outcome |
| CMO1 | When the participants plan their day in the morning | → | they choose their **relief medication** dosage **preventively** | → | which may impact their CSMS for the day |
| CMO2 | When the participants experience allergy symptoms | → | their behaviour changes as they try to **avoid being exposed** to grass pollen | → | which may result in fewer symptoms and less medication reported in the CSMS |
| CMO3 | When the participants **stay either inside or outside** | → | their exposure may differ from the **national pollen count** | → | which may affect the degree of their symptoms reported in the CSMS |
| CMO4 | When the participants assess their symptoms while completing the CSMS | → | they may **mistake other symptoms** for allergy symptoms | → | which may lead to overreporting of symptoms caused by grass pollen in the CSMS |
| CMO5 | When the participants **travel abroad** | → | their exposure to pollen may differ from the **national pollen count** | → | which may affect their CSMS |
| CMO6 | When the participants have symptoms caused by **allergies other than pollen-induced rhinoconjunctivitis** and use medication as a result | → | they still report this use of medication in the CSMS | → | which may **lead to overreporting** of medication for symptoms associated with rhinoconjunctivitis |
| CMO7 | When the participants have allergy symptoms that do not occur in the CSMS | → | they **cannot report all their symptoms** | → | which may lead to underreporting of grass pollen-induced symptoms in the CSMS |

Table 4. Comment distribution.

| Comment distribution | | | | | | | |
|---|---|---|---|---|---|---|---|
| CMO configuration | Total | | | Treatment | | Placebo | |
| | n | comments | median [min;max] | n | median [min;max] | n | median [min;max] |
| CMO1 | 15 | 22 | 1 [1;6] | 10 | 1 [1;6] | 5 | 1 [1;2] |
| CMO2 | 15 | 22 | 1 [1;3] | 12 | 1.5 [1;3] | 3 | 1 [1;1] |
| CMO3 | 141 | 412 | 2 [1;88] | 102 | 2 [1;28] | 39 | 2 [1;88] |
| Inside | 35 | 70 | 1 [1;8] | 27 | 1 [1;8] | 8 | 1 [1;3] |
| Inside (open windows) | 2 | 62 | 31 [7;55] | 1 | 7 | 1 | 55 |
| Outside | 104 | 280 | 2 [1;32] | 74 | 1 [1;18] | 30 | 2 [1;32] |
| CMO4 | 33 | 79 | 1 [1;14] | 23 | 1 [1;14] | 10 | 1 [1;6] |
| CMO5 | 92 | 364 | 2 [1;20] | 63 | 2 [1;20] | 29 | 2 [1;12] |
| CMO6 | 8 | 13 | 1 [1;5] | 7 | 1 [1;5] | 1 | 1 [1;2] |
| CMO7 | 88 | 234 | 2 [1;31] | 58 | 2 [1;31] | 30 | 2 [1;23] |
| Itchy mouth or throat | 29 | 97 | 2 [1;20] | 19 | 2 [1;20] | 10 | 1 [1;6] |
| Mucus in the throat | 14 | 41 | 1 [1;14] | 9 | 1 [1;20] | 5 | 2 [1;10] |
| Headache | 14 | 27 | 1 [1;7] | 10 | 1 [1;7] | 4 | 2.5 [1;3] |
| Swollen eyes | 12 | 19 | 1 [1;4] | 7 | 1 [1;4] | 5 | 1 [1;2] |
| Stinging or burning eyes | 11 | 30 | 1 [1;9] | 8 | 1.5 [1;9] | 3 | 1 [1;3] |
| Respiratory problems | 8 | 20 | 2 [1;5] | 5 | 2 [1;4] | 3 | 1 [1;5] |

**CMO4 – Symptoms.** This CMO configuration was supported by comments about confusing symptoms. For example, some participants were unsure whether their symptoms were caused by their allergy or a cold, which is illustrated by these comments: "I don't know if it's allergies or a cold" and "Difficult to separate allergy from cold!". Another example could be if their symptoms were caused by another allergy, as one participant commented: "I don't think it's pollen but dust or something". This can lead to overreporting of symptoms as participants tend to report all symptoms in the CSMS questionnaire, even if they are unsure whether they are caused by a cold or related to another allergy.

**CMO5 – Travel abroad.** Many participants did not respond to the CSMS questionnaire while abroad because their pollen exposure might differ from the national pollen count: "I'm in Barcelona, where there is no pollen apparently", and "Just came back to DK after a week on Mallorca without symptoms". Others still responded, but their responses may lead to information bias because these answers cannot be related to the national pollen count. Furthermore, it may lead to a lower response rate and, thereby, lower retention: "I have been abroad for the last few weeks and therefore have not reported data".

**CMO6 – Use of medication because of another type of allergy.** The participants' reporting of medication use due to other allergies, such as "Have taken medication due to cat allergy" or "I have taken antihistamine in connection with a hives attack", may affect the CSMS. Medication consumption attributed to rhinoconjunctivitis symptoms may be overestimated. Additionally, symptoms specific to grass pollen-induced ARC may be underestimated because the medication alleviates symptoms from other allergies. Conversely, symptoms may be overestimated if participants report the symptoms associated with other allergies (cf. CMO7).

**CMO7 – Other allergy symptoms.** This CMO configuration was supported by comments where participants reported symptoms that do not occur in the CSMS. The most frequently mentioned symptoms were as follows: itchy mouth or throat, mucus in the throat, headache, swollen eyes, stinging or burning eyes, and respiratory problems. For example, three different participants commented: "Very itchy mouth throughout the day", "Itching and mucus in the throat", and "Headache, mucus, trouble breathing". Not being able to report these allergy symptoms may lead to an underestimation of the symptom score.

 

## Discussion

This realist evaluation of the contexts and mechanisms that may affect the CSMS collected from free text boxes in the ILIT.NU trial synthesized comments from 233 of 404 eligible participants and generated 7 CMO configurations. Key mechanisms for the outcome of the CSMS included CMO1 – decision on preventive relief medication dosage, CMO2 – exhibiting symptom-relieving behaviour, CMO3 – being exposed to different levels of grass pollen, CMO4 – mistaking other symptoms for grass pollen-induced symptoms, CMO5 – different exposure to grass pollen when travelling abroad, CMO6 – reporting of medication for other allergies in CSMS, and CMO7 – failure to report symptoms not included. The seven CMO configurations are discussed below in three overall themes.

### Symptoms and medication

This section includes CMO1, CMO4, CMO6, and CMO7. The analysis showed that at least 33 of participants reported symptoms unrelated to grass pollen-induced ARC, and 8 participants reported use of medication not related to this. This misreporting may lead to overestimation of the CSMS due to participants' symptoms seeming more severe than they are. This may compromise the validity of the CSMS. Validity is defined as '*the degree to which an instrument truly measures the construct(s) it purports to measure*'(13, p. 150). When participants report symptoms and medication use caused by other factors, the CSMS may not truly measure the construct it is intended to address. This misreporting may lead to inaccurate conclusions about the efficacy of ILIT. Hence, it may be important to explore the option of enabling participants to report symptoms resulting from factors not related to grass pollen-induced ARC, and medication taken for unrelated allergies. This will help address reporting issues and ensure the validity of the trial results. However, participants may find it challenging to differentiate between symptoms caused by grass pollen-induced ARC and those that are unrelated. In contrast, if such misreporting occurs evenly across the treatment and placebo groups, it may not impact the trial's outcomes.

Fifteen participants reported using preventive relief medication one to two times during the grass pollen season. The use of preventive medication has implications for the CSMS. Patients who take preventive medication may inadvertently inflate their scores due to unnecessary medication usage. Moreover, this practice complicates the trial design as participants cannot ascertain whether they would have experienced symptoms without this preventive medication. Although participants were instructed not to use relief medication preventively during the ILIT.NU trial, expert opinion encourages participants to do so when not in a clinical trial [27,28]. Some participants may not have followed the instructions in the trial because of this. In future trials, it would be relevant to emphasize to participants the importance of avoiding preventive medication to ensure valid trial results.

Furthermore, if the treatment is effective, participants in the treatment group may have a lower CSMS due to ILIT, while participants in the placebo group may be more inclined to take preventive medication to reduce their symptoms, which may result in a lower CSMS due to the preventive medication. This makes it more challenging to differentiate between the groups and detect the efficacy of ILIT – however, ILIT.NU has addressed this by including the medication dose and the free text boxes.

A total of 88 participants reported symptoms they believed were caused by grass pollen, which were not included in the original CSMS. The most frequently reported symptom was an itchy mouth or throat, experienced by 29 participants. This symptom is common in ARC but is also a recognized side effect of sublingual immunotherapy [29,30]. It's inclusion in the CSMS could complicate results in studies focused on sublingual immunotherapy, but it remains relevant for a comprehensive examination of ARC. Therefore, it should be considered for inclusion in the CSMS for general ARC trials, while excluding it in sublingual immunotherapy studies.

Headache, another symptom identified in CMO7, is a debated symptom in ARC. It is challenging to differentiate between rhinosinusitis headaches and migraines [31], making its integration into the CSMS difficult. Fatigue might be another relevant symptom. We have not covered fatigue in the current analysis, as the participants had the opportunity to report it in the updated CSMS, along with medication dosage and difficulties concentrating. Nevertheless, both the

literature and the participant feedback from the focus groups and the free text boxes indicate that these aspects are relevant and important for the outcome of the CSMS. Therefore, these aspects would be beneficial in future trials, as demonstrated by the ILIT.NU trial, where medication dosage played a crucial role in differentiating groups [20].

Additionally, the free text boxes allowed participants to report other symptoms, which ranged in prevalence from 8 to 14 participants. This highlights the importance of ensuring that participants can report all relevant symptoms, as failure to do so may lead to underestimating the total symptom burden and compromising validity [13]. Conversely, only frequently recognized symptoms must be included; thus, outliers are avoided. Another concern would be whether adding questions may increase the response burden and, thereby, lower the response rate. On the other hand, participants may experience frustration when they are unable to report common symptoms they perceive as crucial.

Given these points, further investigation is needed to assess the potential benefit of incorporating some of these symptoms into the CSMS. An alternative approach could be to include ad hoc questions with a free text box, as used in the ILIT.NU trial, to capture additional symptoms while maintaining the clarity and focus of the CSMS. This may enhance the comprehensiveness of symptom reporting.

### Exposure and behaviour

This section includes CMO2 and CMO3.
Patient comments highlighted the variability in grass pollen exposure, influenced by outdoor activities and time spent indoors. Individual choices regarding exposure, balanced against the individual risk of allergic reactions, further contributed to this variability. By avoiding exposure to grass pollen, participants' symptoms may decrease without medication use. If it is mostly participants in the placebo group who tend to avoid exposure to grass pollen, it may compromise the possibility of detecting a difference between the trial groups. This behaviour could potentially affect the trial outcomes, leading to information bias. Therefore, it would be of interest to detect the wide-ranging variations in pollen exposure experienced by individual participants. Unfortunately, this was not possible in the ILIT.NU trial. Instead, pollen exposure was measured using a single pollen sampler in West Denmark. An EAACI position paper from 2017 cautions that relying on a single pollen sampler may not accurately reflect individual exposure levels, given activity differences [32]. Instead, a personal pollen sampler could be effective, as it would account for the effects of the duration, time, location, and intensity of individuals' activities on personal pollen exposure [32,33]. Although this approach is more expensive, it may reduce the required sample size by providing more accurate data and decreasing measurement errors [34]. Conversely, participants may find it annoying and inconvenient to wear a personal pollen sampler.

An alternative approach to determining participants' exposure involves incorporating questions into the CSMS regarding subjective perceptions of exposure and the duration of outdoor activities. Additionally, an ad hoc question about the intensity of any outdoor activities could be added, as such activities impact exposure [33]. While this information may not independently influence the CSMS, it can enrich the understanding of participants' experiences, treatment, and ARC.

We propose to emphasize the importance of challenging the allergy. Additionally, we consider it important to focus on exposure and inquire about this, as the placebo group may change their behaviour more than the intervention group due to greater symptoms.

### Travel abroad

This section includes CMO5.
The pollen season in Denmark lasts from May to August, which coincides with the summer holidays and a common period of travel abroad. This was also evident in the ILIT.NU trial, where 92 of the participants reported having travelled abroad for varying durations during this period. This may have implications for a trial's validity and results. In the ILIT.NU trial, pollen exposure was measured using the single pollen sampler in West Denmark as a reference point for all participants, regardless of their location. However, pollen levels and types vary between countries, and the impact of pollen exposure

on symptoms can differ based on the region and year [32]. Therefore, if foreign travel is not registered and taken into account, information bias may occur. Secondly, this may lead to a lower response rate, as some participants skip reporting their CSMS while travelling abroad. To address these challenges, we recommend enabling participants to report if they are abroad in their CSMS. This could involve specifying the country of stay to facilitate comparison with local pollen measurements. Alternatively, participants could provide information about the duration of their stay abroad, instead of answering the questionnaire, allowing for adjustments for missing data.

## The realist evaluation approach

Realist evaluation is an approach that acknowledges the complexity of reality and the importance of processes in shaping outcomes. By identifying seven CMOs affecting the CSMS, this study contributes to the growing body of realist evaluations exploring how contextual factors influence patient-reported outcomes.

Malengreaux et al. emphasize that realist evaluation is particularly useful for evaluating complex interventions where outcomes depend on specific contexts [35]. Similarly, our study shows that mechanisms such as adjusting medication intake (CMO1) or misattributing symptoms (CMO4) are shaped by external factors, directly influencing CSMS validity. This supports the realist evaluation's role in understanding "what works, for whom, and in what context." [22,23].

Another key contribution of the realist evaluation is explaining variability in outcomes. Factors such as differences in pollen exposure when traveling (CMO5) and reporting medication use for unrelated allergies (CMO6) illustrate how individual experiences impact patient-reported outcomes. As Malengreaux et al. note, accounting for such variations is crucial in health interventions [35].

While previous studies have emphasized the need for further validation of the CSMS, this study provides new insights by identifying specific contextual mechanisms that may affect its accuracy and applicability [10,11]. These include, for example, the influence of exposure variability (CMO3), reporting of unrelated symptoms (CMO6), and travelling abroad during the pollen season (CMO5). By making these mechanisms explicit, the study contributes to refining our understanding of *why* and *under what circumstances* the CSMS may not function as intended. These insights are grounded in the experiences, reflections, and reported practices of individuals living with ARC, aligning with COSMIN recommendations that only the target population can adequately assess the comprehensibility, relevance, and completeness of a patient-reported outcome measure [8,9]. This offers a practical framework for identifying potential sources of bias, guiding future psychometric validation, and informing targeted adjustments to the CSMS in clinical trials.

Furthermore, their review highlights the importance of stakeholder engagement in realist evaluation. Our study aligns with this by incorporating patient-reported experiences, which provided critical insights into how participants interpreted and reported symptoms and medication use. This reinforces the need for patient involvement in measurement development to improve the validity of patient-reported outcomes [8,9,36,37].

Overall, our findings support existing realist evaluations by demonstrating the importance of contextual influences in patient-reported outcomes. Applying the realist evaluation approach enhances understanding of how measurement tools function in real-world settings and highlights the need for refining patient-reported outcomes to improve accuracy in clinical trials [8,9,35–37].

## Strengths and limitations

This realist evaluation goes beyond assessing the efficacy of allergen-specific immunotherapy for ARC using the CSMS. It explores the contexts and mechanisms that may influence the outcome of the CSMS. The involvement and use of qualitative methods may ensure that the CSMS effectively captures the experiences and symptoms of the participants for whom it is intended [11]. This provides valuable insights into participants' perspectives, enriching the understanding of the complexity of the CSMS. The credibility and reliability of the findings are enhanced because the analysis incorporates data from 233 participants. Given that commenting in the free text boxes was voluntary, it remains uncertain whether

the participants who did not provide comments did not experience any of the CMO configurations or simply chose not to report them. It is possible that a larger number of participants would have contributed to the identified CMOs if the questions had been mandatory. Consequently, a notable limitation of this realist evaluation is that 171 out of the total population of 404 participants chose not to elaborate on their answers in the CSMS, potentially leading to selection bias. Furthermore, if participants had been directly involved in the research process of this evaluation, they might have provided additional perspectives. The absence of an expert group or broader stakeholder involvement in the evaluation process may limit the range of perspectives considered and the insights generated [22,24] However, the results were subject to oversight by qualified researchers, ensuring the rigour and credibility of the research process.

We have provided some recommendations for future research, suggesting ways to explore how to address challenges such as participants travelling abroad, use of preventive medication, reporting unrelated symptoms and medication, varying pollen exposure, and the potential inclusion of additional symptoms in the CSMS. Furthermore, medication dosage, fatigue, and difficulties concentrating are also relevant aspects, as addressed in the ILIT.NU trial. These identified issues could help differentiate between the treatment and placebo groups, preventing information bias as well as overestimation or underestimation of results. However, this realist evaluation only provides a qualitative perspective, and further quantitative studies are needed. There is most evidence for the CMOs about exposure (CMO3), travel abroad (CMO5), and other symptoms (CMO7), where itchy mouth or throat is the most prevalent additional symptom. These CMOs are particularly recommended for further investigation. Additionally, it may be an option to start further research by investigating the responsiveness of the items in the CSMS [38]. We acknowledge the variability of pollen levels and types across countries, recognizing that the same pollen quantity can lead to different symptom levels [32]. Although conducted within a Danish study population, the findings of this realist evaluation likely capture general issues applicable to the CSMS and ARC.

## Conclusion

In this realist evaluation, we have identified and verified contexts and mechanisms that may impact the outcome of the CSMS. Awareness of these CMO configurations may improve the internal validity of future allergy trials using CSMS as an endpoint. Therefore, we recommend that future studies investigate these factors further, including how variability in pollen exposure, preventive medication use, long-distance foreign travel, and reporting of unrelated or additional symptoms may introduce bias. These findings inform potential future adjustments to the CSMS by enabling participants to report contextual information or include relevant symptoms not currently covered. However, such changes would require a new psychometric validation study before being implemented in clinical research.

## Author contributions

**Conceptualization:** Anne Møller Lohmann, Anne Poder Petersen, Johannes Martin Schmid, Hans Jürgen Hoffmann, Jeanette Finderup.

**Data curation:** Anne Møller Lohmann.

**Formal analysis:** Anne Møller Lohmann.

**Investigation:** Anne Møller Lohmann.

**Methodology:** Anne Møller Lohmann, Jeanette Finderup.

**Project administration:** Anne Møller Lohmann.

**Supervision:** Anne Poder Petersen, Johannes Martin Schmid, Hans Jürgen Hoffmann, Jeanette Finderup.

**Validation:** Anne Møller Lohmann, Anne Poder Petersen, Jeanette Finderup.

**Visualization:** Anne Møller Lohmann.

**Writing – original draft:** Anne Møller Lohmann.

**Writing – review & editing:** Anne Møller Lohmann, Anne Poder Petersen, Johannes Martin Schmid, Hans Jürgen Hoffmann, Jeanette Finderup.

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
