## [Decision Letter · Decision Letter 0]

Dear Dr. Lohmann,

Thank you for submitting your manuscript to PLOS ONE. After careful consideration, we feel that it has merit but does not fully meet PLOS ONE’s publication criteria as it currently stands. Therefore, we invite you to submit a revised version of the manuscript that addresses the points raised during the review process.

**ACADEMIC EDITOR:** please make the corrections according to reviewer's comments or write a detailed rebuttal on a point by point basis.

We look forward to receiving your revised manuscript.

Kind regards,

Davor Plavec, MD, MSc, PhD, Prof.

Academic Editor

PLOS ONE

“The ILIT.NU trial, including this realist evaluation, is financed by Innovation Fund Denmark, Aarhus University, and Aarhus University Hospital.”

3. In the online submission form, you indicated that [The data that support the findings of this study are available on request from the sponsor, Hans Jürgen Hoffmann. The data are not publicly available due to privacy or ethical restrictions.].

Additional Editor Comments:

Dear Authors,

please make the corrections according to reviewer's comments or write a detailed rebuttal on a point by point basis.

Reviewers' comments:

Reviewer's Responses to Questions

**Comments to the Author**

1. Is the manuscript technically sound, and do the data support the conclusions?

Reviewer #1: Yes

Reviewer #2: Yes

2. Has the statistical analysis been performed appropriately and rigorously?

Reviewer #1: Yes

Reviewer #2: Yes

3. Have the authors made all data underlying the findings in their manuscript fully available?

Reviewer #1: Yes

Reviewer #2: Yes

4. Is the manuscript presented in an intelligible fashion and written in standard English?

Reviewer #1: Yes

Reviewer #2: Yes

Reviewer #1: Comments

The method of processing information and monitoring the outcome of treatment through questionnaires in patients with chronic disease is an established method, but your review and analysis are important because of the connection of all the information you have collected.

Reviewer #2: Dear Author,

This paper presents a very interesting perspective on CSMS and the validity of the questionnaire.

Could you clarify how you decided on the five preliminary themes? This is not clearly explained in the methods section.

The results section could be written in a more concise manner. Tables 1 and 4 are somewhat repetitive. Additionally, the refinement of the CMO should be described in the methods section. There are many descriptive statistics that could be presented in a table format.

In the discussion section, it would be helpful to include a comparison with similar research to better understand the impact of these CMOs on the final study results.

Overall, I believe this paper could be written in a more engaging way.

**Do you want your identity to be public for this peer review?** For information about this choice, including consent withdrawal, please see our Privacy Policy

Reviewer #1: No

Reviewer #2: No

---

## [Author Response · Author response to Decision Letter 1]

3 Apr 2025

Answer:

We have thoroughly reviewed the templates to confirm that everything adheres to PLOS ONE’s style requirements.

“The ILIT.NU trial, including this realist evaluation, is financed by Innovation Fund Denmark, Aarhus University, and Aarhus University Hospital.”

Answer:

Thank you for bringing it to our attention. We have now deleted the Acknowledgement section and updated the Funding Statement.

The data used for this realist evaluation were collected as part of the ILIT.NU trial. The ILIT.NU trial received funding from Innovation Fund Denmark, Aarhus University and Aarhus University Hospital. However, this specific realist evaluation did not receive any funding.

3. In the online submission form, you indicated that [The data that support the findings of this study are available on request from the sponsor, Hans Jürgen Hoffmann. The data are not publicly available due to privacy or ethical restrictions.].

Answer:

The data supporting the findings of this study contain potentially sensitive information, including health-related data and personally identifiable comments from free text boxes that cannot be fully anonymized. As a result, data sharing is subject to ethical and legal restrictions. The authors do not have sole authority over data access decisions.

Our ethical approval requires that data sharing complies with the European Parliament and Council Regulation No. 2016/679 of 27 April 2016 (General Data Protection Regulation, GDPR). Making the data publicly available would conflict with GDPR requirements, which mandate the protection of personal and sensitive information. Therefore, data access is restricted in accordance with these regulations and the guidelines set by The Scientific Ethical Committees for Region Midtjylland, which has imposed these restrictions to protect participant confidentiality.

Requests for access to the data can be submitted to:

The Scientific Ethical Committees for Region Midtjylland

Skottenborg 26

DK-8800 Viborg

Tel: +45 7841 0183

Email: komite@rm.dk

Website: www.komite.rm.dk

Reviewer #1:

The method of processing information and monitoring the outcome of treatment through questionnaires in patients with chronic disease is an established method, but your review and analysis are important because of the connection of all the information you have collected.

Answer:

Thank you for your positive feedback and for recognizing the significance of this approach. While questionnaire-based outcome monitoring is well established, we appreciate your recognition of our study’s value in integrating qualitative data to provide a comprehensive perspective on the Combined Symptoms Medication Score.

Reviewer #2:

Dear Author,

This paper presents a very interesting perspective on CSMS and the validity of the questionnaire.

Answer:

Thank you.

Could you clarify how you decided on the five preliminary themes? This is not clearly explained in the methods section.

Answer:

The five preliminary themes was developed using an inductive approach. This has been elaborated in the methods section to ensure greater transparency.

The results section could be written in a more concise manner. Tables 1 and 4 are somewhat repetitive. Additionally, the refinement of the CMO should be described in the methods section. There are many descriptive statistics that could be presented in a table format.

Answer:

We have revised the results section to make it more concise and precise. Additionally, we have moved Table 4 up before Table 3 and incorporated the descriptive statistics into the former Table 4 (now Table 3), removing them from the text to improve readability. You are right that Tables 1 and 4 (now Table 3) are somewhat similar; however, we believe it is important to include both to maintain transparency regarding the differences between the preliminary and final CMOs. We appreciate your insightful suggestions.

In the discussion section, it would be helpful to include a comparison with similar research to better understand the impact of these CMOs on the final study results.

Answer:

To compare our findings with similar research, we have addressed this in the discussion section titled "The Realist Evaluation Approach." Here, we compare our results with the systematic review by Malengreaux et al. (2022), which examines the use of realist evaluation in health research. This comparison helps contextualize the impact of our identified CMOs and highlights how contextual influences shape outcome measures in realist evaluations.

Overall, I believe this paper could be written in a more engaging way.

Answer:

We have carefully revised the manuscript, making it more engaging and your suggestion above about including the descriptive statistics moving it from the text, has improved

Best regards,

Anne Møller Lohmann

Department of Public Health – Respiratory Diseases and Allergy,

Aarhus University and Aarhus University Hospital,

Palle Juul-Jensens Boulevard 99,

8200 Aarhus N, Denmark

Email: anneml@clin.au.dk

---

## [Decision Letter · Decision Letter 1]

Dear Dr. Lohmann,

Thank you for submitting your manuscript to PLOS ONE. After careful consideration, we feel that it has merit but does not fully meet PLOS ONE’s publication criteria as it currently stands. Therefore, we invite you to submit a revised version of the manuscript that addresses the points raised during the review process.

We look forward to receiving your revised manuscript.

Kind regards,

Davor Plavec, MD, MSc, PhD, Prof.

Academic Editor

PLOS ONE

Journal Requirements:

Additional Editor Comments:

Dear Authors, please revise your manuscript in accordance with the reviewer's comments or write a detailed rebuttal on a point-by-point basis.

Reviewers' comments:

Reviewer's Responses to Questions

**Comments to the Author**

Reviewer #2: All comments have been addressed

2. Is the manuscript technically sound, and do the data support the conclusions?

Reviewer #2: Yes

3. Has the statistical analysis been performed appropriately and rigorously?

Reviewer #2: Yes

4. Have the authors made all data underlying the findings in their manuscript fully available?

Reviewer #2: Yes

5. Is the manuscript presented in an intelligible fashion and written in standard English?

Reviewer #2: Yes

Reviewer #2: Dear Authors,

Overall, this is a significant improvement. The revised manuscript offers a valuable contribution to understanding the Combined Symptom Medication Score (CSMS) through a realist evaluation framework.

The manuscript now more clearly explains the inductive approach used for deriving preliminary themes. However, here are some additional comments to improve:

1. including a brief example or quote in the methods section illustrating the transition from qualitative data to theme could further support transparency.

2. The comparison with prior realist evaluations (e.g., Malengreaux et al., 2022) is appreciated. The discussion could benefit from a more explicit statement on how this study uniquely contributes to the ongoing refinement of CSMS in clinical research beyond confirming prior concerns.

3. While the study highlights threats to internal validity (e.g., preventive medication use, pollen exposure variability), more direct recommendations for mitigating these issues in future CSMS-based trials would strengthen the conclusion.

4. Consider also providing more visual clarity in the CMO tables—perhaps bolding key terms or restructuring to facilitate readability.

5. A few sections of the discussion—particularly those addressing additional symptoms—might be more concise for improved flow.

6. If available, additional details on the development or piloting of the updated CSMS (with ad hoc questions) could help contextualize its use and interpretation.

With these minor revisions the manuscript will be well suited for publication.

**Do you want your identity to be public for this peer review?** For information about this choice, including consent withdrawal, please see our Privacy Policy

Reviewer #2: No

---

## [Author Response · Author response to Decision Letter 2]

25 May 2025

Reviewer #2:

Dear Author,

Overall, this is a significant improvement. The revised manuscript offers a valuable contribution to understanding the Combined Symptom Medication Score (CSMS) through a realist evaluation framework.

The manuscript now more clearly explains the inductive approach used for deriving preliminary themes. However, here are some additional comments to improve:

Answer:

Thank you for your kind and constructive feedback. We’re pleased that the revisions have clarified our approach, and that the contribution is valuable. We address your additional comments below.

1. including a brief example or quote in the methods section illustrating the transition from qualitative data to theme could further support transparency.

Answer:

Thank you for this helpful suggestion. We have now included a participant quote in the Methods section (Step 1) to illustrate how reflections from the focus groups contributed to the development of a theme. This addition aims to enhance transparency around the transition from qualitative data to theme generation.

2. The comparison with prior realist evaluations (e.g., Malengreaux et al., 2022) is appreciated. The discussion could benefit from a more explicit statement on how this study uniquely contributes to the ongoing refinement of CSMS in clinical research beyond confirming prior concerns.

Answer:

Thank you for this valuable feedback. We have revised the discussion to more clearly state how the study uniquely contributes to the refinement of the CSMS. Specifically, we now highlight how the findings reveal concrete contextual mechanisms—based on the experiences and reflections of individuals with ARC—that may affect the CSMS’s accuracy and applicability. This aligns with COSMIN recommendations and offers a framework for guiding future validation and refinement efforts.

3. While the study highlights threats to internal validity (e.g., preventive medication use, pollen exposure variability), more direct recommendations for mitigating these issues in future CSMS-based trials would strengthen the conclusion.

Answer:

We have revised the conclusion to more clearly reflect how future studies may explore ways to address the identified contextual factors that can affect the CSMS—such as variability in pollen exposure, use of preventive medication, foreign travel, and the reporting of unrelated or additional symptoms. We now outline how such findings may inform potential future adjustments to the CSMS. However, we have taken care not to propose direct modifications to the instrument itself, as any such changes would require a new psychometric validation study.

4. Consider also providing more visual clarity in the CMO tables—perhaps bolding key terms or restructuring to facilitate readability.

Answer:

Thank you for this helpful suggestion. As recommended, we have improved the visual clarity of the CMO tables by bolding the most important elements to facilitate easier reading and navigation of key information.

5. A few sections of the discussion—particularly those addressing additional symptoms—might be more concise for improved flow.

Answer:

Thank you for the suggestion. We have revised the section on additional symptoms to make it more concise and improve the overall flow of the discussion.

6. If available, additional details on the development or piloting of the updated CSMS (with ad hoc questions) could help contextualize its use and interpretation.

Answer:

Thank you for the comment. We have added a brief description of how the updated CSMS was developed based on input from focus groups with participants in the ILIT.NU trial. We hope this contextualizes its use and interpretation.

Best regards,

Anne Møller Lohmann

Department of Public Health – Respiratory Diseases and Allergy,

Aarhus University and Aarhus University Hospital,

Palle Juul-Jensens Boulevard 99,

8200 Aarhus N, Denmark

Email: anneml@clin.au.dk

---

## [Editor Report · Decision Letter 2]

Understanding the Combined Symptom Medication Score in the light of contexts and mechanisms

PONE-D-24-49594R2

Dear Dr. Lohmann,

We’re pleased to inform you that your manuscript has been judged scientifically suitable for publication and will be formally accepted for publication once it meets all outstanding technical requirements.

Kind regards,

Davor Plavec, MD, MSc, PhD, Prof.

Academic Editor

PLOS ONE

Additional Editor Comments (optional):

Dear Authors, your manuscript is now ready for publication in it's current form.
---

## [Editor Report · Acceptance letter]

PONE-D-24-49594R2

PLOS ONE

Dear Dr. Lohmann,

I'm pleased to inform you that your manuscript has been deemed suitable for publication in PLOS ONE. Congratulations! Your manuscript is now being handed over to our production team.

Kind regards,

on behalf of

Dr. Davor Plavec

Academic Editor

PLOS ONE